# The Influence of Regulatory Focus on Media Choice in Interpersonal Conflicts

**Stefan Tretter \*** and **Sarah Diefenbach**

Department Psychology, Ludwig Maximilian University of Munich, Leopoldstr. 13, 80802 Munich, Germany; sarah.diefenbach@lmu.de
\* Correspondence: stefan.tretter@lmu.de; Tel.: +49-89-2180-6368

**Abstract:** People's choices of (electronic) communication channels are central to the quality of communication—and sometimes detrimental to their actual communication goals. However, while factors influencing media choice are abundant, potential means to intentionally influence these choices are scarce within computer-mediated communication research. We explore the role of regulatory focus as one possible factor to understand and influence media choice in interpersonal conflicts. Regulatory focus theory proposes two motivational systems, promotion (i.e., needs for nurturance and growth) and prevention (i.e., needs for safety and security), that account for differences in preferred strategies for goal-pursuit. In a vignette-based study, we manipulated the situational regulatory focus (promotion or prevention) and surveyed participants' preferred media choice for a hypothetical conflict scenario. Our results show that the induction of a dominant prevention focus (vs. promotion focus) leads to a shift in preference towards leaner communication media and channels that establish a higher subjective buffer between sender and receiver (e.g., text-messaging over calling). We elaborate on how these findings contribute to the understanding of media choice in interpersonal conflicts and point out potential ways to influence behavior through the design of communication technologies. Limitations of the present study and future research opportunities are discussed.

**Keywords:** media choice; regulatory focus theory; buffer effect; interpersonal conflicts; computer-mediated communication

## 1. Introduction

More than ever, communication nowadays takes place via a plethora of devices and services. Organizations allow and encourage their employees more and more to work from home, and a majority of private communication already happens within digital contexts. Especially in the light of a world that has been hit by the covid-19 pandemic, resulting in lockdowns and social distancing, face-to-face conversations have been increasingly replaced by computer-mediated communication (CMC). This puts even more emphasis on the appropriate choices of communication media for such purposes that previously might have been addressed in person, since communication itself and its outcomes can be significantly affected by the channel it funnels through [1]. For example, without the additional information that can be derived from vocal intonation, a simple text message like "Can you call me asap?" might be a cause of worry or excitement [2].

While CMC research has extensively studied what media people should choose when, as well as what media they do choose and why, theoretically recommended and actual choices do not always coincide. In order to bridge this gap, it would be relevant to identify potential adjusting factors to influence media choices without changing the cornerstones of a given situation. Such knowledge could pave the way towards means to deliberately elicit beneficial media choices as suggested by pertinent theories and empirical research. For example, given managers' anecdotes about how (especially younger) employees'

aversion to calling others can pose a threat to sales and recruitment [3], there should be a growing interest in how to increase people's willingness to pick up the phone when it is beneficial to the task at hand.

Although discrepancies between actual choices and theory-based recommendations occur regularly, this does not imply that people do not follow plausible motives underlying their decisions. In the face of interpersonal conflicts, for instance, it surely seems appealing to avoid direct confrontation through media choice, e.g., by leaving someone a text message about a critical issue rather than talking to them directly on the phone. After all, to shield oneself from the receiver and thereby "buffer" negative experiences is one common reason to use technology instead of communicating face-to-face [4–6]. But as attractive as this way of conduct might seem at times, the appeal of lean media bears the risk of being detrimental to the communication itself. For example, people are overconfident about their ability to interpret emotions in emails [7] and might misinterpret their content more negatively than intended by the sender [8]. Those pitfalls would in turn render the use of rich media more beneficial to the accurate exchange of emotions and successful conflict resolution.

Since interpersonal conflicts constitute situations of opposing motives, i.e., avoiding short-term negative experiences through lean media vs. approaching long-term solutions through rich media, they provide an appropriate application field to explore ways to guide people's media choice behavior. To this end, we build on the well-established psychological concept of regulatory focus and draw a line between the motivational orientation in critical communication situations and how it affects media choice. By consciously inducing a situationally dominant regulatory focus, we intend to influence people's media preferences when confronted with an interpersonal conflict situation. Given the propositions of earlier research about the effectiveness of different media for certain communication goals, this approach could furthermore pave the way towards more elaborate means to support beneficial media choices.

## 2. Theoretical Background

### 2.1. Media Choice

#### 2.1.1. Uses, Gratifications and Motives

How people choose between channels for communication has been a long-standing question in CMC research [9–11]. Early theories of CMC, first and foremost media richness theory [12], were dedicated to explaining effectiveness, i.e., what media characteristics allow for best performances under which circumstances. Soon, they were also applied to predict which media people actually choose. However, this approach often yielded contradicting evidence [10,13,14], since people tend to not act strictly rational but also according to their subjective needs. The so-called uses and gratifications approach (U&G), originally developed with mass media in mind, addresses this by focusing on the individual needs that are sought to be gratified by engaging with a certain medium [15–17]. Accordingly, predictions of media choice become more reliable when we understand why people use certain channels and how they decide between the options available to them. On the other hand, this enables us to create communication media that address the central motivational orientations people adhere to.

#### 2.1.2. Media as Means to an End

In general, each communicative act serves instrumental, self-presentational, and relational goals [18] to varying degrees and people use media in order to pursue these goals. In many everyday instances, people choose their communication channels quite pragmatically (e.g., what is accessible, easy to use or suits individual preferences). But beyond that, especially in cases of sensitive communication subjects, other needs can come to the fore and the media choice may vary accordingly [19,20]. Simply speaking, people may anticipate how they will feel if they communicate the same message via different media, and will choose the medium that promises the best feelings for themselves—thereby following the basic hedonic principle to approach pleasure and avoid pain [21]. In other words,

they strive towards desired end-states and move away from undesired end-states [22]. As such, media choice can serve as a means to take control over the communication process and the personal emotional outcomes.

In general, the hedonic principle as a driver of media choice becomes especially relevant in high-stakes situations with an inherent possibility of negative emotional consequences [23], as typically associated with the conveyance of negative messages. For positive messages, media choice is less of an issue since the message is less ambiguous and easier to interpret, while negative messages can be stressful for the sender as well as the receiver [4]. People anticipate the undesirable effect that a negative message might have for them or their relationship and adapt their transmission accordingly [24]. That is why it may be tempting to convey such messages via technological means since their mediating nature helps to insulate the sender from the probably unpleasant feedback of the receiver [5]. This capability of communication media to shield oneself from others' reactions when communicating critical content has sometimes been referred to as the "buffer effect" of media and has been reported by several authors [5,6,25,26].

Note, however, that choosing communication media to avoid direct confrontation is not the only way to deal with the communication of critical content. Instead of employing a buffer effect, one might utilize media choice to take control over a potentially threatening conversation in other ways. For example, people differ in their perceived ability to use a channel to express themselves as intended [27] and might be more confident to soften the impact of a message on the receiver face-to-face [6]. They may even prefer talking to someone face-to-face over calling them since it provides a more accurate assessment of how it affects the other [24].

Taken together, previous research shows that the communication channel an individual prefers is highly context- and subject-dependent [28], and media choice is a means to take control over the communication process in the desired way. Specifically, one can vary the emotional intensity and interactional speed in socio-emotional contexts via the chosen communication channel [29]. For instance, email as an a-synchronous and text-based communication medium promotes slower exchange of messages and lesser emotional cues than the telephone. Thus, the former provides more room for reflection and controlled answers than the latter.

In the face of interpersonal conflicts, this opportunity to control the upcoming communication process is likely to have substantial consequences for its outcomes. One's motive underlying media choice might either be to avoid the conflict and prevent escalation or to approach the conflict and strive for a resolution. That is why it is relevant to understand the psychological processes that regulate behavior in such critical situations and how they could be consciously influenced. The theory of regulatory focus provides a promising concept in this endeavor.

### 2.2. Regulatory Focus Theory

### 2.2.1. Outcomes and Strategies

Regulatory focus theory posits that humans possess two motivational systems that are rooted in fundamental needs and regulate their behavior: promotion and prevention [21]. The promotion system is based on needs for nurturance and growth, while the prevention system is based on needs for safety and security [30]. Consequently, people with a promotion focus are more sensitive to the presence and absence of positive outcomes, i.e., gains and non-gains, while people with a prevention focus are mainly concerned with the presence and absence of negative outcomes, i.e., losses and non-losses [31]. This also affects how people experience the status-quo. When there is no change of situation from one time to another, prevention-focused individuals would consider this as a success, since the situation did not become worse (a non-loss). Conversely, promotion-focused individuals would see this as a missed chance to improve the situation (a non-gain), therefore considering it a failure [32]. Notably, these systems are independent, i.e., a person can be

high or low on both at the same time, and while one might have a predominant disposition, regulatory focus is also affected by one's current situation [33].

But people do not only differ in their conceptualization of desired and undesired end-states, but also with regard to the preferred strategic means they employ to pursue their goals [22]. Promotion focus regulation involves a preference for eager, advancement strategies and promotes approach behaviors, while prevention focus regulation, in contrast, leads to a preference for vigilant, cautious strategies that elicit avoidance behaviors [21]. Therefore, "a person who wants to get a good grade on a quiz (a desired end-state), for example, could either study hard at the library the day before the quiz (approaching a match to the desired end-state) or turn down an invitation to go out drinking with friends the night before the quiz (avoiding a mis-match to the desired end-state)" [34] (p. 117). In sum, there can be different paths to the same goal, but preference towards a particular path is determined by individuals' regulatory focus.

### 2.2.2. Regulatory Focus in Conflicts

As outlined above, media choice gains importance in negatively-valenced situations, particularly when the issue is not just a threat to each parties' subjective well-being but their relationship [4]. This becomes especially important when the source of negativity not only pertains to the actual act of sharing negative information but resides in a potential disagreement between both parties. Interpersonal conflicts are usually grounded in some kind of incompatibility and are one of the most common stressors in daily life. But they are not exclusively negative since conflicts can also contribute to a deeper understanding of oneself and the affected relationship [35]. Thus, interpersonal conflicts are particularly suited to investigate regulatory focus' role in media choice because they bear the burden of emotional intensity and uncertain outcomes. However, which channel senders choose to handle them may well be affected by their currently dominant motivation to pursue a vigilant, avoiding or eager, approaching strategy.

An investigation of how people deal with critical and potentially conflict-evoking situations can profit from regulatory focus theory in two ways: By considering how people tend to conceptualize its possible outcomes on the one hand and, on the other hand, what strategic means they prefer. First, considering the uncertain result of a conflict, the subjective probability of a positive or negative outcome can be affected by regulatory focus. In anticipation of future events, people with a prevention focus tend to prefer pessimistic forecasts, whereas those with a promotion focus show a preference towards optimistic forecasts [36]. Furthermore, research has shown that promotion-focused individuals perceive demanding tasks, in our case interpersonal conflicts, more as a challenge than a threat compared to their prevention-focused counterparts [37].

Second, aside from the different expectations of outcomes, potential conflicts also provide the opportunity to be handled in different ways. For example, people primed with a promotion focus display more risk-seeking behaviors, while an induced prevention focus leads to risk avoidance [38]. In price negotiations, usually a communication situation with conflicting goals, prevention-focused individuals prefer vigilant, loss-minimizing strategies, whereas promotion-focused individuals prefer eager, gain-maximizing strategies [39]. Also, Rodrigues et al. [40] report that in relationships, partners with a prevention focus tend towards conflict avoidance, while a promotion focus is associated with more conflict solution strategies.

Taken together, people vary in their perception of outcomes and their preference for certain strategies depending on their currently dominant regulatory focus. Accordingly, we suppose that different media choices also appear more or less suited to handle an interpersonal conflict due to their differences in richness and buffer effect. In a promotion focus, people may see the opportunity for conflict solution through direct confrontation and thus prefer richer media, while in a prevention focus, people may tend towards conflict-avoidance and prefer leaner media with a higher buffering effect. Therefore, the active

manipulation of individuals' focus may influence people's attitude towards available options and consequently their media choice.

*2.3. The Present Study*

In line with previous research outlined above, we assume that people anticipate the course of an upcoming conflict and its consequences and take this into consideration when choosing among media. While some people might have more confidence in face-to-face conversations or technological means that closely resemble it, there is evidence for an increasing preference towards CMC in cases of negative messages [5,24,25]. This can be attributed to the subjective buffer effect of media which constitutes a metaphorical shield that provides a feeling of control over the interaction and safety from aversive reactions. While communication media differ in their objective characteristics and therefore their subjective buffer effects, we suppose that people's preference towards higher buffering media vary depending on their currently dominant regulatory focus. More precisely, prevention focus is related to pessimistic anticipation, loss-minimizing behavior as well as risk and conflict avoidance. Thus, we presume that in the face of interpersonal conflicts, prevention-focused individuals tend to choose media with a higher buffering effect than promotion-focused individuals.

**Hypothesis 1 (H1).** *Prevention (vs. promotion) focus leads to the choice of channels with a higher subjective buffer for the communication about interpersonal conflicts.*

Although people differ in their perception of communication media [27,41,42] and a channel's buffering effect is one particular subjective media characteristic [5,26], this individual variance is limited. There are still objective characteristics that have led to a common understanding of a media richness continuum on which different channels can be arrayed along [43,44]. According to the widely used media richness theory, channels vary in their richness due to differences regarding the speed of interaction, the multiplicity of cues, the language variety, and the personal focus a channel establishes [45]. For example, the telephone is considered richer than written text [46], and email leaner than voicemail [10]. In a broad sense, these characteristics are closely intertwined with the modality a channel uses, i.e., if it is text- or speech-based, therefore sending fewer cues, and interactional speed, i.e., if it is synchronous or asynchronous, therefore providing slower feedback [47]. Accordingly, we categorize media in the following order from leanest to richest: text-based and a-synchronous, text-based and synchronous, speech-based and a-synchronous, speech-based and synchronous. Note that this order based on the concept of richness is also in parallel to other conceptualizations of prominent media theories, like social presence [11,44] or media synchronicity [13,48]. Those theories would suggest an identical order, with text-based and a-synchronous channels at one end to speech-based and synchronous channels at the other end of the social presence and synchronicity continuum, respectively. In addition to H1, which relies on the subjective perception of a medium, namely its buffer effect, we assume that people will also show differences in their actual choice of media.

**Hypothesis 2 (H2).** *Prevention (vs. promotion) focus leads to a higher probability of choosing leaner channels for the communication about interpersonal conflicts.*

We conducted a vignette study to test our assumptions regarding the influence of regulatory focus on the choice of media for interpersonal conflicts. By priming different regulatory foci, our study, for one thing, aims to contribute to a better understanding of channel preferences in conflicts, but most notably explores a way to influence media choice in otherwise identical situations.

## 3. Materials and Methods

### 3.1. Design

In order to test the assumptions regarding the influence of regulatory focus on media choice for interpersonal conflicts, we conducted a vignette-based online study where participants were randomly assigned to one of two regulatory focus manipulations. They were either primed with a promotion or prevention focus and subsequently indicated their medium of choice for a potential conflict situation. Afterward, they also rated the chosen medium according to its subjective buffering effect. In addition, chronic regulatory focus and interpersonal closeness were assessed to control for potentially confounding variables.

### 3.2. Materials

3.2.1. Chronic Regulatory Focus

The German Regulatory Focus Questionnaire (RFQ) [49] was applied to control for chronic differences in individuals' regulatory focus. The German RFQ assesses the general regulatory focus orientation with eleven items on a five-point scale ranging from "never or seldom"/"never true"/"certainly false" (1) to "very often"/"very often true"/"certainly true" (5). It consists of six items for promotion focus, e.g., "How often have you accomplished things that got you 'psyched' to work even harder?", and five items for prevention focus, e.g., "Not being careful enough has gotten me into trouble at times.", partly reverse-coded. Internal consistencies for the promotion and prevention focus scales of the German RFQ were acceptable with a Cronbach's $\alpha$ of 0.725 and 0.797, respectively (see Table 1 for additional descriptive statistics).

**Table 1.** Psychometric Properties of Inclusion of Other in the Self Scale (IOS), German Regulatory Focus Questionnaire (RFQ) and Buffer Scores.

| Scale | M | SD | Min | Max | Cronbach's $\alpha$ |
|---|---|---|---|---|---|
| Interpersonal Closeness (IOS) | 3.25 | 1.82 | 1.00 | 7.00 | n/a |
| Regulatory Focus (RFQ) | | | | | |
|   Promotion Score | 3.50 | 0.61 | 2.00 | 4.67 | 0.725 |
|   Prevention Score | 3.32 | 0.72 | 2.00 | 5.00 | 0.797 |
| Channel Buffer Score | | | | | |
|   text-based and a-synchronous (e.g., email, SMS) | 3.96 | 0.70 | 3.00 | 5.00 | 0.784 |
|   text-based and synchronous (e.g., chat, instant messenger) | 3.56 | 0.54 | 2.83 | 4.33 | 0.727 |
|   speech-based and a-synchronous (e.g., voice message, voicemail) | 3.11 | 0.99 | 2.00 | 4.83 | 0.648 |
|   speech-based and synchronous (e.g., telephone) | 2.41 | 0.75 | 1.00 | 4.00 | 0.775 |

3.2.2. Regulatory Focus Induction

To manipulate regulatory focus, we adapted a well-established approach [50] where people are asked to think and write about either their ideals and hopes, i.e., inducing promotion focus, or duties and obligations, i.e., inducing prevention focus (note: to enhance the effect, we also asked participants to list strategies to fulfill these goals, a common method to induce regulatory fit, but implemented no non-fit condition; see limitations section for further discussion). Within the promotion condition people encountered the following task description:

> "Think about an aspiration, a hope, or an ideal that you currently hold and want to accomplish. It should be something at which you want to have success. Afterward, list three things you can do to most possibly succeed in that."

On the other hand, within the prevention condition, participants were presented with the following instruction:

"Think about a duty, responsibility, or obligation that you currently hold and have to fulfill. It should be something at which you should not make mistakes. Afterward, list three things you can do to most possibly not fail in that."

In both conditions, participants described the goals they thought about and listed three ways to pursue them.

### 3.2.3. Vignette

Participants were asked to put themselves in a scenario revolving around a potential conflict and briefly describe their imagined situation. The vignette closely followed the structure from O' Sullivan [5] and read:

"Imagine a situation, in which you are about to communicate with another person. This should be about an issue that could lead to a conflict between you and this person. This potential conflict could result from

- you having different opinions about a topic,
- you doing something the other person considers unacceptable,
- or you having critique, that might hurt the other person.

Please shortly describe the concrete situation that you are imagining."

### 3.2.4. Media Choice

Participants were asked to "Imagine you are in the situation described before, which way of communication would you prefer?". They had four options of media to choose between, clustered among modality and synchronicity. The possible answers included "written and not synchronous (e.g., email, SMS)", "written and synchronous (e.g., chat, instant messenger)", "spoken and not synchronous (e.g., voice message, voicemail)", or "spoken and synchronous (e.g., telephone)".

### 3.2.5. Interpersonal Closeness

Since the relationship to the receiver has been shown to play a pivotal role in the use of media [19,25], the Inclusion of Other in the Self Scale (IOS) was used as a lean and reliable measure for interpersonal closeness [51,52]. By selecting the appropriate pair of increasingly overlapping circles, representing the persons involved, people indicate how close they feel to the respective other on a pictorial seven-point scale. Detailed statistics regarding the IOS are presented in Table 1.

### 3.2.6. Buffer Score

The subjective buffering effect of a medium, i.e., the capacity to establish a metaphorical shield between sender and receiver, was measured with six items adapted from Wotipka [26]. On a scale from "strongly disagree" (1) to "strongly agree" (5), people assessed the respective medium with items like "This channel makes me feel like I am protected from the reactions of the other person." or "This channel offers me protection to say what I want to say." Internal consistencies were acceptable for three of the four possible media choices: text-based and a-synchronous (Cronbach's $\alpha$ = 0.784), text-based and synchronous (Cronbach's $\alpha$ = 0.727), speech-based and synchronous (Cronbach's $\alpha$ = 0.775). The scale to assess the subjective buffering effect of speech-based, asynchronous media, however, yielded a questionable Cronbach's $\alpha$ of 0.648. Nevertheless, we refrained from deleting one item that would have led to an improvement to maintain consistency across media assessments (see Table 1 for detailed statistics).

### 3.2.7. Attention Check

In order to control for participants' attention and engagement in the study, we included a dummy question which participants were instructed to *not* answer. More specifically, participants have been presented the following instruction towards the end of the questionnaire: "We are interested in whether you have taken the time to thoroughly read the

instructions and understand them correctly. To demonstrate that you have read and understood the instructions, please ignore the following question and click 'proceed'. Thank you." The subsequent (dummy) single-choice question asked for the individual's most preferred communication medium. Participants who answered this question probably did not read the instructions attentively and thus failed the attention check.

*3.3. Procedure*

After confirming an initial consent agreement, participants filled out the German RFQ and afterward were presented with either the promotion or prevention focus induction. They wrote down one of their current hopes/ideals or duties/obligations and three ways they intend to pursue them. This was disguised as a way to guarantee their engagement in the study. Subsequently, participants were asked to put themselves in the outlined potential conflict situation and shortly describe what situation they concretely imagined. Afterward, they indicated the communication channel they would choose and rated the relationship with the imagined receiver on the IOS. Following that, participants rated the respective media on the buffer scale, were asked to report their age as well as gender, and encountered the actual attention check described above. Finally, they were thanked and received instructions on how to acquire their compensation.

*3.4. Sample*

The study's final sample consisted of 80 participants (59% male, 40% female, 1% diverse) with a mean age of M = 35.5 (SD = 11.5; Min = 18; Max = 70). Initially, 140 participants were recruited via Clickworker, a German-based crowd-working platform similar to Amazon Mechanical Turk and received EUR 1.20 as compensation for their participation. The initial sample size was based on a priori power analysis with a pre-defined alpha level of 0.05 and a power of 0.80. Since regulatory focus induction is at the center of our study design, we retrieved the lowest reported effect size, i.e., Cohen's d = 0.56, from Freitas and Higgins [50] who applied a similar manipulation. This yielded a recommended sample size of 104 participants.

Participants had to be at an age of 18 to 99 years and speak German as a first language. Those who failed the simple attention check, i.e., who answered the dummy question (43% of the initial sample), were excluded from further analyses (see Section 3.2.7. Attention Check). This resulted in a remaining sample of 80 participants. Besides, we also performed a manipulation check regarding the regulatory focus inducing writing task, verifying that participants actually wrote about aspirations, hopes or ideals (in the promotion condition) or duties, responsibilities or obligations (in the prevention condition). The manipulation check was positive for all remaining participants, resulting in the final sample size of 80 (see Supplementary Materials for the open data set).

## 4. Results

As an initial analysis, we explored the relationship between different communication media and their assigned buffer scores, assuming that media with those characteristics that label a medium as rich should in turn lead to less buffering experiences. The decline in buffer score from text-based, a-synchronous to speech-based, synchronous channels depicted in Table 1 is supported by a significant statistical relationship between medium and buffer score according to Spearman's rank-order correlation ($r_s = -0.586$, $p < 0.001$). This supports the implicitly presumed interplay between the dependent variables underlying our hypotheses. While H1 explores the influence of regulatory focus on differences in subjective buffer scores of the chosen media, H2 considers the concrete media choice and their ranking along the richness continuum.

In line with H1, participants chose communication channels with higher subjective buffer scores for the interpersonal conflict if they previously encountered a prevention focus induction (N = 42, M = 2.95, SD = 1.00) compared to a promotion focus induction (N = 38, M = 2.54, SD = 0.80; t(78) = 2.043, $p = 0.044$, Cohen's d = 0.45). A subsequent

ANCOVA controlling for each participant's chronic promotion and prevention focus as well as the respective interpersonal closeness as covariates yielded analogous results (F(1,75) = 4.083, $p$ = 0.047, partial $\eta^2$ = 0.052).

Furthermore, to test the assumption of H2 that people in a prevention focus tend to choose leaner media than people in a promotion focus, we applied an ordinal logistic regression with regulatory focus as the predictor and actual media choice as the criterion. Overall, the model showed a good fit to the data, since neither the Pearson goodness-of-fit test ($\chi^2$(2) = 1.057, $p$ = 0.529) nor the deviance goodness-of-fit test ($\chi^2$(2) = 1.097, $p$ = 0.549) yielded significant results. Furthermore, the model predicted media choice significantly better than the intercept-only model ($\chi^2$(1) = 5.923, $p$ = 0.015). The odds of choosing a leaner medium rose by a factor of 3.487 if people were in a prevention compared to a promotion focus, which constitutes a statistically significant effect ($\chi^2$(1) = 5.386, $p$ = 0.020). Thus, since the odds ratio is larger than one, the probability of choosing a leaner medium increases for prevention-focused individuals compared to promotion-focus individuals, which is in line with H2. This effect holds true when chronic regulatory focus scores and interpersonal closeness are incorporated into the regression (OR = 3.546; $\chi^2$(1) = 5.432, $p$ = 0.020), while their integration even leads to a poorer model fit compared to the intercept-only model ($\chi^2$(4) = 7.117, $p$ = 0.130). Detailed statistics for both logistic regression models are presented in Table 2.

**Table 2.** Logistic regression results for the prediction of media choice [1].

| Model | Variable | B [5] | SE [6] | Wald $\chi^2$ | $p$ | OR [7] | 95%-CI OR | |
|---|---|---|---|---|---|---|---|---|
| | | | | | | | Lower | Upper |
| **Model I** [2] | Regulatory Focus Manipulation [4] | 1.25 | 0.54 | 5.39 | 0.020 | 3.49 | 1.21 | 10.01 |
| **Model II** [3] | Regulatory Focus Manipulation [4] | 1.27 | 0.54 | 5.43 | 0.020 | 3.55 | 1.22 | 10.28 |
| | Chronic Promotion Focus Score | −0.34 | 0.42 | 0.67 | 0.412 | 0.71 | 0.31 | 1.61 |
| | Chronic Prevention Focus Score | 0.19 | 0.34 | 0.33 | 0.566 | 1.21 | 0.63 | 2.35 |
| | Interpersonal Closeness | 0.10 | 0.14 | 0.47 | 0.491 | 1.10 | 0.83 | 1.46 |

[1] Higher scores indicating leaner channels. [2] $R^2$ Nagelkerke = 0.085. [3] $R^2$ Nagelkerke = 0.102. [4] Regulatory Focus Coding: Promotion = 0, Prevention = 1. [5] Unstandardized Regression Coefficient. [6] Standard Error. [7] Odds Ratio.

## 5. Discussion

Plenty of works in the field of CMC research have shown that media choice is dependent on a variety of individual differences, e.g., personality [53], experience [54], or competence [55], as well as contextual factors, e.g., message valence [6], receiver [25], or culture [56]. However, within those studies, these factors are either pre-determined by the sender's traits or essential components of the situation are varied. We explored a way to influence people's media choices without changing anything about the situation itself, by conducting an a-priori manipulation and even controlling for sender-specific (chronic regulatory focus) and receiver-related (relationship) variables. We did this by adapting the propositions of regulatory focus theory to a new application field, namely media choice in interpersonal conflicts. We were able to show that prevention focus induction led to a preference for media with higher buffering effects and, more importantly, to an actual shift in probability of choosing leaner communication channels compared to a promotion focus induction.

Of course, this insight is yet limited to the particular case of interpersonal conflicts and even within those conflicts, it is not a given which communication medium constitutes the best choice. For example, media richness theory would propose different media depending on whether a conflict is based on an absence of information, i.e., uncertainty, or multiple interpretations of a situation, i.e., ambiguity [14]. Similarly, media synchronicity theory would characterize appropriate media choices based on the required processes, whether new information has to be transmitted and processed, i.e., conveyance, or a mutual understanding of known information is to be established, i.e., convergence [13,57].

Nevertheless, while the identification of beneficial media choices has been and always will be a subject central to CMC research, we laid out a new approach to nudge people accordingly. It is up to future investigations, to what extent this approach can be applied to other communication situations and which other ways to influence people's media choice might also prove effective.

### 5.1. Theoretical Implications

Our work provides several theoretical contributions to research regarding media choice, regulatory focus, and interpersonal conflicts. First, our study outlines the potential influence of regulatory focus on media choice. Prominent theories in the field of CMC center on adequate choices of communication channels in terms of better performance. However, in order to predict actual media choices, it is more important to understand why people choose a certain channel, a notion central to the uses and gratifications approach. Regarding conflict situations, we referred to the buffer effect of media as a means to cope with the upcoming conversation. By choosing leaner media, people might seek to satisfy an active motive to prevent unpleasant experiences and avoid direct confrontation. On the other hand, people might see richer media as an opportunity to better tackle the roots of a conflict. Our results reveal that people show different preferences depending on the activated regulatory focus and support the assumption that accompanying approach and avoidance motives play a role in media choice. Thus, regulatory focus theory can contribute to the understanding of the underlying psychological processes of media choice that take place when distressing situations might harm individuals' well-being as well as their relationships.

Second, taking the conflict research outlined above into account, our results are in line with empirical evidence regarding the association between regulatory focus and behavioral tendencies. Furthermore, these associations might provide alternative or complementing explanations for our observations apart from the particular buffer of a medium. Using a channel's buffer effect to shield oneself from negative reactions of a receiver can be considered as a self-serving behavior, valuing own needs over those of others. Similarly, Winterheld and Simpson [58] reported that in romantic relationships people with a prevention focus perceive their partners as more distancing and less supportive than with a promotion focus. Moreover, prevention-focused individuals approached conflicts by discussing details of the conflict while promotion-focused individuals displayed more creative conflict solution strategies. These tendencies correspond to the choice of communication technology observed by us, since leaner media, i.e., text-based and/or a-synchronous, allow for a better elaboration of a conflict's details and their processing, while richer media, i.e., speech-based and synchronous, allow for a back-and-forth and emotional displays, that enable better discussions about the possible resolutions and ways out of a disagreement.

This link between a communicational strategy and certain media characteristics can also be found in the preference for accuracy over speed tactics depending on regulatory focus [33,59]. In a prevention focus, people express a stronger preference for accuracy, which might be addressed by choosing a-synchronous media. On the other hand, promotion-focused individuals prioritize speed, which is why they might prefer faster communication channels to immediately resolve conflicts. This link between regulatory focus and media choice could also be drawn on a more basic cognitive processing level, insofar as prevention focus is associated with local processing of information and promotion focus with global processing [60]. Local processing of a dispute might result in a stronger focus on the content and circumstances of a conflict, while global processing might be more strongly represented in the intent to maintain the relationship. Each way of processing would in turn suggest a different communication channel that fits the respective individual's priorities.

Last, our work contributes to the study of media choice by taking focus away from the prediction of media choice to its deliberate manipulation. To our knowledge, this approach is the first attempt to influence media choice by manipulating participants' situational

regulatory focus. While there might be associations with people's chronic regulatory focus and the preference for certain media [61], our approach enabled us to influence media choice without interfering with the cornerstones of a communicational act, that is sender, receiver, and message. We did not vary the incident, an interpersonal conflict, and controlled for individual differences as well as the sender–receiver relationship. Yet, we were able to elicit differences in people's preferences towards communication channels by conducting a preceding regulatory focus manipulation. If this intervention would be for better or worse in real-life scenarios, as mentioned above, might depend on the particular circumstances and is open to further inquiry, but it extends the possibilities of media choice research with an opportunity to do so.

*5.2. Practical Implications*

As much as the opportunity to influence media choice by varying regulatory focus contributes to research, it indicates potential ways to foster better communication in everyday life. Practical applications of regulatory focus theory can be found in a wide range of contexts such as work [62,63], health [64,65], or consumer behavior [66,67]. Similarly, even though the effects of a certain dominant focus and the corresponding media choice on communication itself were not covered by our study design, regulatory focus theory could potentially be utilized to support successful conflict communication and long-lasting relationships. The use of text-messaging instead of face-to-face communication, for example, has been associated with an increase in distancing behavior in couples' conflicts [68]. Moreover, people who highly value interpersonal relationships anticipate more negative consequences and show higher tendencies for conflict avoidance [69,70]. At the same time, avoidance-oriented individuals are prone to exhibit negative communication behaviors that might harm overall relationship satisfaction [71]. This indicates that choices of lean media, as well as avoidance-motivated behavior, can be detrimental to the resolution of conflicts and thereby might bring relationships in jeopardy. Our insights suggest that a deliberate promotion focus induction could counteract these behavioral tendencies by increasing individuals' preferences for richer media and at the same time promote a more beneficial conflict approach.

Of course, within real-life scenarios, an experimental manipulation of regulatory focus as applied within our study seems impractical. However, the present study could still support conflict management by applying our insights on the role of regulatory focus in media choice to the design of communication technologies. While we demonstrated that the manipulation of regulatory focus impacts media choice, media itself could, in turn, be designed in a way that it not only appeals to people with a particular regulatory focus but actually induces it. Taking this thought one step further, such a deliberate induction through design may also foster beneficial communication behaviors in all situations where research suggests a particularly preferable regulatory focus.

Given the notion that media itself could support interpersonal communication by affecting regulatory focus, the question that arises is how to organically implement such a manipulation within real-life situations. Unfortunately, feasible and realistic ways to manipulate regulatory focus are still to be found since research in the field mostly employs verbal priming methods, whereas non-verbal methods would provide a more applicable approach to the design of media. For example, a more subtle and implicit way to induce a certain regulatory focus than the approach we choose by asking participants to write about their goals and ways to achieve them was used by R. S. Friedman and Förster [72]. They applied a pictorial maze task that participants had to solve in advance. This task was either framed in promotion terms, i.e., a mouse trying to find to a piece of cheese, or prevention terms, i.e., a mouse seeking shelter from an owl. Approaches like these seem to provide a more economic and unobtrusive way to change regulatory focus than completing a writing task. However, this kind of non-verbal manipulation is still separated from the actual application and does not provide a solution to naturally integrate the

desired manipulation. In fact, the literature on non-verbal, integrated regulatory focus induction is still scarce.

The most promising way to achieve such an implicit manipulation would be to implement visual cues that are psychologically associated with either approach or avoidance motivation. For instance, Mehta and Zhu [73] reported several studies supporting the relationship of the color red with avoidance and blue with approach motivation and their influence on performance of different tasks. Similarly, Elliot et al. [74] also report evidence for a link between red and avoidance motivation. Furthermore, neuroimaging studies by Bar and Neta [75] show higher activation of the amygdala, which is associated with fear processing, when people were presented with sharp objects compared to their counterparts with a curved contour. Although pertinent research is still inconclusive [76], such insights on the association of visual characteristics with changes in cognitive processing might contribute to the design of user interfaces and its effect on users' strategic orientation. For example, avoiding red and sharply shaped design elements for communication media might obviate the activation of a prevention focus, while blue elements and roundly shaped edges might even foster a promotion focus. This could in turn affect the choice and usage of communication media for interpersonal conflicts. Nevertheless, this is highly speculative (and does not apply to voice-only channels, e.g., phones) and should be a subject—among others—of future research inquiries.

*5.3. Limitations and Future Research*

Especially since the present study applied a rather new approach by bringing regulatory focus theory and media choice together, some limitations have to be discussed. First, while our initial sample exceeded our intended sample size by far, this was no longer the case after we excluded participants due to failed attention checks. This led to a lower statistical power, and while we still found a significant effect on media choice, a larger sample size would indisputably yield more reliable estimates of effect sizes.

Second, there are restrictions inherent to the application of a hypothetical scenario to which participants responded to. Though vignette studies are a common and recognized method to investigate phenomena under controlled circumstances [77], the question about whether participants' answers correspond to their actual behavior in real life conditions definitely calls for further investigation. On the one hand, people can be subject to a social desirability bias [78], indicating media choices they think they ought to choose. For example, one might believe that rich channels are the only appropriate way to handle conflicts since they convey a symbolic message of goodwill to resolve the conflict [79]. On the other hand, people might be mistaken about their behavior given that they actually experience the described distressing episode, since self-assessments might be a good, but definitely not perfect predictor of actual behavior [80,81].

Third, there are limitations concerning the particular content of the presented vignette. We provided several examples of potential conflicts emerging from disagreements between sender and receiver to make sure participants can recall or imagine a relatable situation. More specifically, we asked participants to imagine conflicts that either routed in different opinions between those involved, social transgressions that the other might condemn or critique towards the receiver. Since these different examples left room for variation among individuals, upcoming research should adhere to more concretely and extensively formulated descriptions to guarantee a shared interpretation of scenarios between all participants. Such situations should involve the concrete cause of the interpersonal conflict as well as the particular relationship between the two parties, in order to examine potential effects of the inherent type of conflict while controlling for potentially confounding factors.

Fourth, future research may profit from the application of systematically varied vignettes, particularly with regard to differences in the emotional and instrumental spectrum. We referred to situations of interpersonal conflicts since their negative valence was expected to induce distress and in turn deliberate considerations about the pros- and cons of available communication channels. We did this because previous research suggests

that the most desirable characteristic for media choice is media accessibility and its ease of use [19], but more profound reasons are taken into account when the message holds a certain socio-emotional valence, i.e., it is positive or negative [5,6]. However, we initially focused on negative scenarios and did not incorporate positive emotions. And even within the negative spectrum, distinguishable emotions like anger and fear might induce different processing of conflict situations [82]. For example, since fear is associated with less control over an event than anger, one might be more receptible to the buffering effects of mediated channels. These variations of emotional aspects in combination with regulatory focus should be part of future media choice research. The same applies to a more multifaceted investigation of communication goals, since communication not only serves self-presentational and relational but also instrumental purposes [18]. For example, some conflicts might require the persuasion of another party or the negotiation of new terms. Such content-related goals, apart from the relational aspect of a conflict, can also play a role in media choice [83] and should be taken into account for further inquiry.

Last, there are two promising starting points for future research that are more focused on the application of regulatory focus theory. As already discussed above, one of them is the exploration of methods to implicitly induce an intended regulatory focus by designing elements and characteristics of communication media accordingly. The other pertains to the well-established phenomenon of regulatory fit. Plenty of research has shown wide-ranging effects of experiential value when one's current regulatory orientation and means of goal pursuit are in line, coming from the experience of "feeling right" [31,33,59]. For example, promotion-focused participants are willing to pay a higher price for an object they choose with an eager strategy, i.e., thinking about what they would gain if they chose it, instead of a vigilant strategy, i.e., what they would lose if they did not choose it. In turn, the opposite observation was made for prevention-focused participants [22]. Similarly, participants with a predominant promotion focus are more persuaded by messages framed in terms of eager means than vigilant means, while the reverse was true for participants with a predominant prevention focus [84]. Accordingly, given our reported association between regulatory focus and media choice, people might experience conflicts differently and value the outcomes more, depending on whether the used communication channel fits their current regulatory focus.

In this context, it should be noted that we applied a writing task typically used to manipulate regulatory focus by asking for obligations (i.e., prevention goals) or aspirations (i.e., promotion goals). But furthermore, we also asked to list strategies to not fail (i.e., vigilant strategies) in the prevention condition and to succeed (i.e., eager strategies) in the promotion condition. This listing of strategies is usually used to induce regulator fit or non-fit by asking for either compatible (promotion-eager and prevention-vigilant) or non-compatible (promotion-vigilant and prevention-eager) strategies to the previously stated goals [50]. In our study, we had no interest in inducing incompatible states and applied a manipulation in which regulatory fit would be assumed in both conditions, thereby balancing possible effects. However, a more precise regulatory focus manipulation by just asking for goals could have been conducted, which represents the final limitation of our study.

## 6. Conclusions

The question of why people choose communication media as well as how they handle interpersonal conflicts is an ongoing challenge of scientific inquiry. Communication media as means to outline and discuss these conflicts play a vital role in their outcome, since the channel itself inevitably affects communication processes. The present research contributes to these branches of research by applying regulatory focus theory in order to understand the psychological underpinnings of media choice and, furthermore, influence behavioral tendencies in such situations. We were able to show that the induction of a prevention focus, compared to a promotion focus, increases people's susceptibility to channels with a higher buffering effect and shifts their preferences towards leaner media. Among other

implications, regulatory focus manipulations might prove to be a way to deliberately influence media choice without changing the cornerstones of a given interpersonal conflict—an endeavor barely represented in current research. In conclusion, our study adds to current media choice and the regulatory focus literature by bringing a well-established motivational theory to an application field of everyday relevance.

**Supplementary Materials:** The data are available online at https://doi.org/10.17605/OSF.IO/YWVE9.

**Author Contributions:** Conceptualization, S.T. and S.D.; methodology, S.T.; validation, S.T. and S.D.; formal analysis, S.T.; investigation, S.T.; resources, S.D.; data curation, S.T.; writing—original draft preparation, S.T.; writing—review and editing, S.D. and S.T.; visualization, S.T.; supervision, S.D.; project administration, S.T. All authors have read and agreed to the published version of the manuscript.

**Funding:** This research received no external funding.

**Institutional Review Board Statement:** The study was conducted according to the guidelines of the Declaration of Helsinki, and approved by the Ethics Committee of LMU Munich's Department of Psychology (protocol code: 15_Tretter_b; date of approval: 08.06.2020).

**Informed Consent Statement:** Informed consent was obtained from all subjects involved in the study.

**Data Availability Statement:** The data presented in this study are openly available in the Open Science Framework (osf.io) at 10.17605/OSF.IO/YWVE9, reference number YWVE9.

**Acknowledgments:** We would like to thank the anonymous reviewers for their greatly helpful comments.

**Conflicts of Interest:** The authors declare no conflict of interest.

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
