# Peer review of "The Influence of Regulatory Focus on Media Choice in Interpersonal Conflicts"

_psych, doi:10.3390/psych3010001_

Round 1
Reviewer 1 Report
Research is well conduced and well documented. I only have some minor suggestions to improve data analysis and research design description. In my opinion, the manuscript is well written (although authors should review the entire document for grammatical mistakes), properly documented, justified and organized, results are well presented and conclusions are directly derived from the results.
Before accepting it for publication I would like the authors to address some minor concerns. I am making mostly minor remarks and it is my hope that they will help to improve the manuscript.
- Please, provide a more detailed description of the selection of the sample. Concretely, what characteristics should have the participants to be recruited via Clickwoker (i.e. gender, age, etc.).
- Please, explain in what consisted the attention check.
- Please, explain why after excluding some participants, additional participants were not contacted to obtain the recommended sample size (104 participants).
- Psychometric properties of the measures used should be reported in the “materials” description not in the results.
- Tables are insufficient to describe the results. I would advise to include a new table with the logistic regression data, adding R Nagelkerke. It will allow to better understand logistic regression data.
- Authors should more clearly discuss the limitations of the present study.
Reviewer 2 Report
Overall, this research was interesting, timely, and theoretically well-grounded in the regulatory focus literature. It is well-written. It was a pleasure to read and review.
Applying chronic measures while manipulating the orientational focus was well designed. The use of lean versus rich communication sources, with the 'buffer' as the underlying explanatory mechanism is interesting.
Applying the results to suggest that one's orientation could be manipulated, thus influencing channel choice is managerially relevant.
Below are some questions where clarification would be beneficial.
- How were subjects assigned to the manipulation treatments?
- Would a frequency analysis of communication channel chosen in a 2x2 diagram be informative, categorized by chronic and manipulated RF states? (N for each category?)
- Was there a manipulation check for the writing exercise?
- How does regulatory fit play into the results? (2X2 may help illuminate this).
- In a 2x2 - RF fit would predict rich channel communication for promotion chronic and manipulated, and 'lean' for prevention chronic and manipulated. Did this turn out to be true?
- What may be interesting to examine would be the 'misfit' frequencies of choice - the 'cross' diagonals...I'm chronic promotion...but given the 'conflict' situation being manipulated as preventative....do people shift?
- (And vice versa - I'm preventative chronic - yet when primed for 'promotion' orientation...do the frequencies of choice shift to richer communication channels?
